# Pan-cancer characterization of ncRNA synergistic competition uncovers potential carcinogenic biomarkers

**Junpeng Zhang**[1]*, **Lin Liu**[2], **Xuemei Wei**[1], **Chunwen Zhao**[1], **Sijing Li**[1], **Jiuyong Li**[2], **Thuc Duy Le**[2]*

**1** School of Engineering, Dali University, Dali, Yunnan, People's Republic of China, **2** UniSA STEM, University of South Australia, Mawson Lakes, South Australia, Australia

* zhangjunpeng411@gmail.com (JZ); thuc.le@unisa.edu.au (TDL)

**Data Availability Statement:** All data supporting the findings of the current study are listed in the SCOM framework (https://github.com/zhangjunpeng411/SCOM/) and the web-based

## Abstract

Non-coding RNAs (ncRNAs) act as important modulators of gene expression and they have been confirmed to play critical roles in the physiology and development of malignant tumors. Understanding the synergism of multiple ncRNAs in competing endogenous RNA (ceRNA) regulation can provide important insights into the mechanisms of malignant tumors caused by ncRNA regulation. In this work, we present a framework, SCOM, for identifying ncRNA synergistic competition. We systematically construct the landscape of ncRNA synergistic competition across 31 malignant tumors, and reveal that malignant tumors tend to share hub ncRNAs rather than the ncRNA interactions involved in the synergistic competition. In addition, the synergistic competition ncRNAs (i.e. ncRNAs involved in the synergistic competition) are likely to be involved in drug resistance, contribute to distinguishing molecular subtypes of malignant tumors, and participate in immune regulation. Furthermore, SCOM can help to infer ncRNA synergistic competition across malignant tumors and uncover potential diagnostic and prognostic biomarkers of malignant tumors. Altogether, the SCOM framework (https://github.com/zhangjunpeng411/SCOM/) and the resulting web-based database SCOMdb (https://comblab.cn/SCOMdb/) serve as a useful resource for exploring ncRNA regulation and to accelerate the identification of carcinogenic biomarkers.

## Author summary

Abundant evidence reveals that ncRNAs are important modulators of gene expression, and it is important for us to understand the regulation of ncRNAs in malignant tumors. In this work, we hypothesize that ncRNAs acting as ceRNAs can crosstalk with each other in the form of synergistic competition. We name the hypothesis as the *ncRNA synergistic competition hypothesis*. To uncover potential carcinogenic biomarkers, we present a framework, SCOM, for identifying ncRNA synergistic competition in malignant tumors. From the perspective of synergistic competition, we have shown that the synergistic competition ncRNAs are likely to involve in drug resistance, contribute to distinguishing molecular subtypes of malignant tumors, and participate in immune regulation. In

resource (https://comblab.cn/SCOMdb/). All other relevant data are within the manuscript and its Supporting Information files.

**Funding:** JZ was supported by the National Natural Science Foundation of China (Grant Number: 61963001), the Yunnan Fundamental Research Projects (Grant Number: 202001AT070024). CZ was supported by the Yunnan Fundamental Research Projects (Grant Number: 202101BA070001-221). TDL was supported by the ARC DECRA (Grant Number: DE200100200). The funders had no role in study design, data collection and analysis, decision to publish, or preparation of the manuscript.

**Competing interests:** The authors have declared that no competing interests exist.

addition, the synergistic competition ncRNAs are potential diagnostic and prognostic biomarkers of malignant tumors. We believe that the proposed framework SCOM and the web-based database SCOMdb can contribute to the design of reliable diagnosis and treatment biomarkers for malignant tumors.

## Introduction

Malignant tumors are a type of complex human diseases characterized by uncontrolled growth and spread of malignant cells and tissue infiltration [1]. According to the latest global cancer statistics from International Agency for Research on Cancer (IARC), malignant tumors are the second leading cause of death worldwide, and severely threaten human life and health [2]. Depending on the type of malignant tumor cells, malignant tumors are mainly divided into two categories: cancer and sarcoma [3]. Compared with healthy cells, the structure and function of genes in tumor cells have changed. It is found that polygenic synergy is one of the key characteristics of malignant tumors [4], suggesting that malignant tumors are usually polygenic disorders and their pathogenesis is caused by multi-gene synergy.

Non-coding RNAs (ncRNAs) are a class of various RNA transcripts that usually cannot be translated into proteins. Ordinarily, ncRNAs are categorized into two types: regulatory ncRNAs and structural ncRNAs [5]. For example, microRNAs (miRNAs), long non-coding RNAs (lncRNAs), circular RNAs (circRNAs) and pseudogenes belong to the type of regulatory ncRNAs, while ribosome RNAs (rRNAs) and transfer RNAs (tRNAs) are considered as structural ncRNAs. Existing studies [6,7] have demonstrated that ncRNAs are closely associated with the occurrence and development of malignant tumors, which opens a promising avenue for the diagnosis and preventive treatment of malignant tumors.

As regulators, ncRNAs can modulate the expression of thousands of genes in the manner of trans-acting and cis-acting [8], thereby driving cell biological responses and fates [7]. In addition to the role of regulators, by soaking up miRNAs, ncRNAs can act as competing endogenous RNAs (ceRNAs) [9] which compete with the target messenger RNAs (mRNAs) of the soaked miRNAs. In the ceRNA context, miRNAs serve as mediators to connect non-coding and coding RNAs, resulting in the ceRNA network (consisting of ceRNAs and mRNAs), a more comprehensive regulatory network than the miRNA regulatory network [10]. So far, three types of ncRNAs, including lncRNAs, circRNAs and pseudogenes have been widely known to function as ceRNAs [11], and are regarded as potential carcinogenic biomarkers in clinical applications [12,13]. For example, lncRNA *H19* serves as a ceRNA for *LIN28* to promote breast cancer stem cell maintenance [14], circRNA *circTP63* functions as a ceRNA for *FOXM1* to accelerate lung squamous cell carcinoma progression [15], and pseudogene *PTENP1* acts as a ceRNA for *PTEN* in prostate cancer [16].

The ceRNA regulation of malignant tumors involves the interactions between multiple non-coding and coding genes. In the ceRNA network in which ncRNAs are involved, the competitive relationships between ncRNAs and target mRNAs are not one-to-one, but many-to-many. For example, previous studies [17–25] have demonstrated that the three ncRNAs, *H19*, *MALAT1* and *PVT1* synergistically compete with the four target genes, *KLF4*, *VEGFA*, *ZEB1* and *ZEB2*. In this work, we call such collective competition by a group of ncRNAs against a group of mRNAs as "ncRNA synergistic competition". It is observed that such complicated gene regulation carries out important tasks in physiological and pathological processes, and is considered to be a key component of cellular regulatory networks [7]. Furthermore, the entire cell system presents the characteristics of functional modularity, and each gene functional

module (i.e. a group of genes interacting or crosstalking with each other frequently and not by chance) is responsible for specific biological functions in complex human diseases, including malignant tumors [26]. These findings indicate that ncRNAs, acting as ceRNAs, may synergistically rather than just independently play important roles in malignant tumors.

In this work, to uncover potential carcinogenic biomarkers, we present the SCOM (Synergistic COMpetition) framework to characterize ncRNA synergistic competition in Pan-cancer. SCOM first systematically predicts ncRNA-related ceRNA networks across 31 tumor types (containing 8001 tumor samples). Based on the proposed *ncRNA synergistic competition hypothesis*, SCOM applies three criteria, (i) significant synergistic competition for mRNAs, (ii) significant positive correlation, and (iii) significant sensitive correlation conditioning on the mRNAs synergistically competed against by the ncRNAs, to predict the ncRNA synergistic competition network (consisting of ncRNAs acting as ceRNAs) from the gene expression data of ncRNAs and target mRNAs and the predicted ncRNA-related ceRNA networks. By using SCOM, we provide a landscape of ncRNA synergistic competition in Pan-cancer, contributing to the discovery of potential carcinogenic biomarkers. Ultimately, SCOM provides a framework for discovering the relationships between the synergistic competition ncRNAs (i.e. ncRNAs involved in the synergistic competition) and malignant tumors, and can help to elucidate ncRNA synergistic competition mechanisms in malignant tumors.

## Materials and methods

### Synergistic COMpetition (SCOM)

The proposed SCOM framework is based on the *ncRNA synergistic competition hypothesis* (**Fig 1A**). For soaking up miRNAs, a pool of ncRNA transcripts (acting as ceRNAs) competes with mRNAs through miRNA response elements (MREs). As a result, the competitive interactions between ncRNAs and mRNAs form a ceRNA network (consisting of ncRNAs acting as ceRNAs and target mRNAs). In the ceRNA network, we have observed that the competitive relationships between ncRNAs and mRNAs are not one-to-one but many-to-many, revealing synergistic competition of ncRNAs. Based on this observation, we hypothesize that ncRNAs acting as ceRNAs can crosstalk with each other in the form of synergistic competition (i.e. ncRNAs synergistically compete with mRNAs). We name the hypothesis as the *ncRNA synergistic competition hypothesis*.

The first step of SCOM (**Fig 1B**) is to predict ncRNA-related ceRNA network of each malignant tumor type. The matched ncRNA (including miRNA, lncRNA and pseudogene) and mRNA expression data in Pan-cancer (consisting of 8001 tumor samples) are obtained from The Cancer Genome Atlas (TCGA) [28] project. The information of miRNA targets is obtained by integrating five well-known databases, including miRTarBase v9.0 [29], TarBase v8.0 [30], LncBase v2.0 [31], NPInter 4.0 [32] and ENCORI [33]. By integrating gene (including ncRNAs and mRNAs) expression data and putative miRNA-target interactions, SCOM extends SPONGE [27] (a three-criterion evaluation method) to predict ncRNA-related ceRNA networks across 31 malignant tumors. The three-criterion evaluation method predicts ceRNA interactions by integrating sample-matched ncRNA (miRNA, lncRNA and pseudogene) and mRNA expression data, and putative miRNA-target interactions. We reason that if a candidate ncRNA-mRNA pair meets the three criteria (significant sharing of miRNAs, significantly positive correlation, and significantly sensitive correlation conditioning on shared miRNAs) with $p$-value $< 0.05$, the candidate ncRNA-mRNA pair is regarded as a ceRNA interaction. All of the predicted ceRNA interactions form a large-scale ncRNA-related ceRNA network, with genes (including lncRNAs, pseudogenes and mRNAs) denoted as nodes and the ncRNA-mRNA competition relationships as undirected edges.

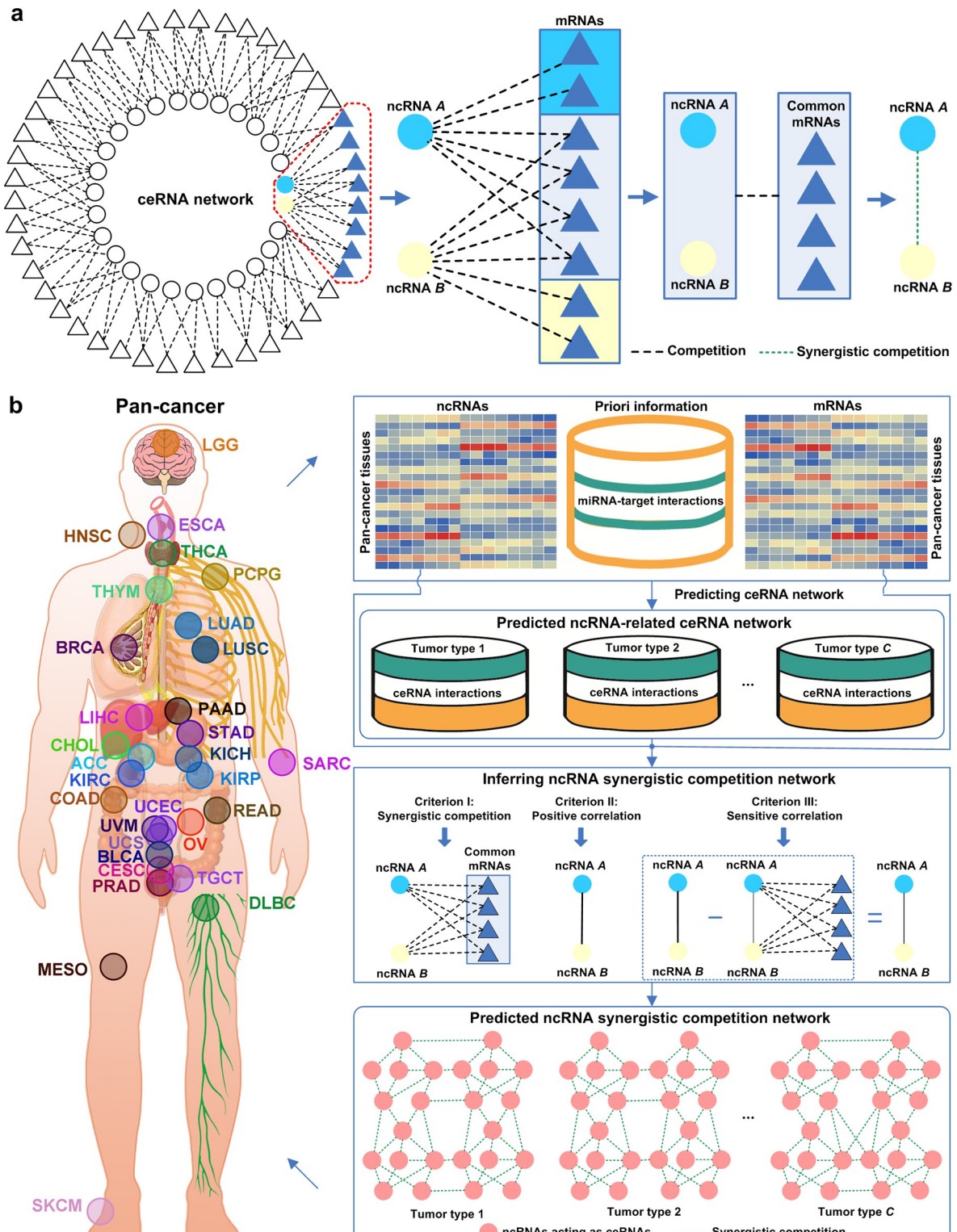

**Fig 1. Schematic illustration of SCOM.** (a) An illustration of the ncRNA synergistic competition hypothesis. ncRNA *A* and ncRNA *B* acting as ceRNAs collectively compete with mRNAs for soaking up miRNAs. In the context of ceRNA competition, the crosstalk between ncRNA *A* and ncRNA *B* is in the form of synergistic competition. (b) For each tumor type in Pan-cancer, SCOM firstly extends the SPONGE [27] method to predict ncRNA-related (including lncRNA- and pseudogene-related) ceRNA network by incorporating gene (including miRNAs, lncRNAs, pseudogenes, and mRNAs) expression data in matched tumor tissues and priori information of miRNA-

target (including miRNA-lncRNA, miRNA-pseudogene and miRNA-mRNA) interactions. By integrating gene (including lncRNAs and pseudogenes acting as ceRNAs, and target mRNAs) expression data in matched tumor tissues and predicted ncRNA-related ceRNA networks, SCOM utilise three criteria to infer ncRNA synergistic competition network for each tumor type in Pan-cancer. Based on the predicted ncRNA synergistic competition networks across 31 malignant tumors, SCOM further uncovers potential carcinogenic biomarkers.

The second step of SCOM (**Fig 1B**) is to infer ncRNA synergistic competition network of each malignant tumor type. For a malignant tumor type, to predict the ncRNA synergistic competition network, we use the expression data of ncRNAs acting as ceRNAs and target mRNAs and the predicted ncRNA-related ceRNA networks. Following the *ncRNA synergistic competition hypothesis*, we consider a ncRNA-ncRNA pair having a synergistic competition relationship if the pair have (i) significant synergistic competition with mRNAs, (ii) significant positive correlation, and (iii) significant sensitive correlation conditioning on their competing mRNAs. The synergistic competition network of a malignant tumor type is then formed by integrating all of ncRNA synergistic competition relationships.

## Pan-cancer transcriptomics data

After removing non-matching tumor samples between ncRNAs (including miRNAs, lncRNAs and pseudogenes) and mRNA expression data, the Pan-cancer transcriptomics data includes the expression data of 17,212 ncRNAs (including 894 miRNAs, 4366 lncRNAs, and 11,952 pseudogenes) and 19,068 mRNAs in 8001 matched tumor samples covering 31 tumor types (**S1 Data**), and the clinical information of the tumor samples (**S2 Data**).

## Putative miRNA-target interactions

Three types of putative miRNA-target interactions, including miRNA-mRNA, miRNA-lncRNA, and miRNA-pseudogene are used for predicting ncRNA-related ceRNA networks. The putative miRNA-mRNA interactions are acquired from miRTarBase v9.0 [29] and Tar-Base v8.0 [30], the putative miRNA-lncRNA interactions are obtained from LncBase v2.0 [31] and NPInter 4.0 [32], and the putative miRNA-pseudogene interactions are from ENCORI [33] (the previous version is starBase). In total, we have 762,540 miRNA-mRNA interactions, 225,063 miRNA-lncRNA interactions, and 91,619 miRNA-pseudogene interactions as priori information of miRNA-target interactions.

## Predicting ceRNA networks in Pan-cancer

Given the gene expression data of miRNAs, lncRNAs, pseudogenes, and mRNAs, and miRNA-target interactions, the above three criteria are used to predict ceRNA network (see **Supplementary Methods** in **S1 File**). In this work, a candidate ncRNA-mRNA pair meeting the three criteria with $p$-value $< 0.05$ is regarded as a ceRNA interaction. For each of the 31 tumor types, we integrate all of ceRNA interactions to form the ceRNA network of the tumor type.

## Constructing ncRNA synergistic competition networks in Pan-cancer

Given the gene expression data of lncRNAs and pseudogenes acting as ceRNAs, and miRNA targets mRNAs and the predicted ncRNA-related ceRNA networks, we use three criteria to construct ncRNA synergistic competition network.

The first criterion uses a hypergeometric test to assess the significance of synergistic competition of ncRNA $i$ and ncRNA $j$ for mRNAs. The significance $p$-value of the test is computed

as:

$$p_{ij} = 1 - \sum_{x_{ij}=0}^{L_{ij}-1} \frac{\binom{M_{ij}}{x_{ij}}\binom{N_{ij}-M_{ij}}{K_{ij}-x_{ij}}}{\binom{N_{ij}}{K_{ij}}} \tag{1}$$

where $N_{ij}$ is the number of all mRNAs in a Pan-cancer dataset, $M_{ij}$ and $K_{ij}$ represent the total numbers of mRNAs competing with ncRNA $i$ and ncRNA $j$ respectively, and $L_{ij}$ is the number of mRNAs synergistically competed by ncRNA $i$ and ncRNA $j$.

The second criterion requires that ncRNA $i$ and ncRNA $j$ are positively correlated. The Pearson [34] method is also utilized to evaluate whether the expression level of ncRNA $i$ and the expression level of ncRNA $j$ are positively correlated at a significant level (e.g. 0.05). The correlation $cor_{ij}$ between ncRNA $i$ and ncRNA $j$ is computed as:

$$cor_{ij} = cor(p,q) = \frac{\sum_{k=1}^{s}(p_k-\bar{p})(q_k-\bar{q})}{\sqrt{\sum_{k=1}^{s}(p_k-\bar{p})^2}\sqrt{\sum_{k=1}^{s}(q_k-\bar{q})^2}} \tag{2}$$

where $p = (p_1,p_2,\ldots,p_s)$ and $q = (q_1,q_2,\ldots,q_s)$ denote the expression level of ncRNA $i$ and ncRNA $j$ respectively, $\bar{p}$ and $\bar{q}$ represent the average expression level of ncRNA $i$ and ncRNA $j$ respectively, and $s$ is the number of samples for each tumor type. Accordingly, the significance $p$-value $pc_{ij}$ is calculated as:

$$pc_{ij} = 2pt\left(cor_{ij}\sqrt{\frac{s-2}{1-(cor_{ij})^2}}\right) \tag{3}$$

where the $pt$ function is used to approximately calculate the probability of $cor_{ij}\sqrt{\frac{s-2}{1-(cor_{ij})^2}}$.

The third criterion is used to evaluate the influence of synergistically competed mRNAs on ncRNA $i$ and ncRNA $j$. The sensitive correlation [35] is also employed to calculate the influence of the mRNAs synergistically competed by ncRNA $i$ and ncRNA $j$, and the null model [27] is also applied to evaluate the significance of this influence. The sensitive correlation $sc_{ij}$ is calculated as:

$$sc_{ij} = cor_{ij} - pcor_{ij} \tag{4}$$

where $pcor_{ij}$ is the partial correlation between ncRNA $i$ and ncRNA $j$, i.e. the correlation conditioning on the mRNAs synergistically competed by the ncRNAs, which is the correlation between ncRNA $i$ and ncRNA $j$ when the influence of the synergistically competed mRNAs is eliminated. $pcor_{ij}$ is calculated as:

$$\begin{aligned} pcor_{ij} &= cor(p,q|M) = cor(p,q|(M_1,M_2,\ldots,M_n)) \\ &= \frac{cor(p,q|(M_1,M_2,\ldots,M_{n-1})) - cor(p,M_n|(M_1,M_2,\ldots,M_{n-1}))cor(q,M_n|(M_1,M_2,\ldots,M_{n-1}))}{\sqrt{1-cor(p,M_n|(M_1,M_2,\ldots,M_{n-1}))^2}\sqrt{1-cor(q,M_n|(M_1,M_2,\ldots,M_{n-1}))^2}} \end{aligned} \tag{5}$$

where $p$, $q$ and $M$ denote the expression level of ncRNA $i$, ncRNA $j$, and the synergistically competed $n$ mRNAs respectively, $cor(p,q|(M_1,M_2,\ldots,M_{n-1}))$ denotes the partial correlation between $p$ and $q$

conditioning on $(M_1, M_2, \ldots, M_n)$, $cor(p,q|(M_1, M_2, \ldots, M_{n-1}))$ represents the partial correlation between $p$ and $q$ conditioning on $(M_1, M_2, \ldots, M_{n-1})$, $cor(p, M_n|(M_1, M_2, \ldots, M_{n-1}))$ is the partial correlation between $p$ and $M_n$ conditioning on $(M_1, M_2, \ldots, M_{n-1})$, $cor(q, M_n|(M_1, M_2, \ldots, M_{n-1}))$ is the partial correlation between $q$ and $M_n$ conditioning on $(M_1, M_2, \ldots, M_{n-1})$.

To evaluate the significance of $sc_{ij}$, the null model hypothesizes that the synergistically competed mRNAs do not affect the correlation between ncRNA $i$ and ncRNA $j$, i.e. the sensitive correlation (the difference between $cor_{ij}$ and $pcor_{ij}$) between ncRNA $i$ and ncRNA $j$ is 0. The number of datasets sampled is set to 1E+03 for the null model, and the pre-computed covariance matrices in the R package SPONGE [27] are used to build the null model. Based on the constructed null model, the significance $p$-value of $sc_{ij}$ is calculated.

In this study, a candidate ncRNA-ncRNA pair meeting the three criteria for synergistic competition with $p$-value < 0.05 is considered as a ncRNA synergistic competition relationship. For each tumor type, we combine all of the ncRNA synergistic competition relationships to create the ncRNA synergistic competition network for the tumor type.

## Network topological analysis

Understanding the topological properties of biological networks, such as node degree, paths and network density can lead to useful insights into the networks. Being scale-free is a property of a complex network (e.g. biological network), and it is determined by the underlying mechanism (e.g. preferential attachment) of a biological network [36]. If the node degree of a biological network obeys a power law distribution, the biological network is regarded as a scale-free network, and this is an important feature of biological networks in the real world [37]. The path and density features are used to measure the tightness of a biological network. Compared with the random networks, a shorter characteristic path length or higher density of a biological network means that the biological network is more likely to be a small-world network [38,39].

The topological features (degree, path and density) of ncRNA-related ceRNA networks and ncRNA synergistic competition networks are analyzed using the R package igraph [40], a useful tool for network analysis and visualization. The ncRNA-related ceRNA networks and ncRNA synergistic competition networks are undirected networks, and the degree of a node is the number of edges connected with the node. The Kolmogorov-Smirnov (KS) test [41] is used to evaluate whether the node degree of a network follows a power law distribution. If the $p$-value of the KS test is smaller than a cutoff (e.g. 0.05), the node degree of a ncRNA-related ceRNA network or a ncRNA synergistic competition network does not follow a power law distribution, then these indicate that the ncRNA-related ceRNA network or the ncRNA synergistic competition network is not a scale-free network. To determine whether the ncRNA-related ceRNA networks and the ncRNA synergistic competition networks are small-world networks, we generate 100 random networks (the same number of nodes and edges with real network) for each ncRNA-related ceRNA network or ncRNA synergistic competition network. The Student's $t$-test [42] is applied to evaluate whether the characteristic path length (or density) of a ncRNA-related ceRNA network or a ncRNA synergistic competition network is smaller (or higher) than that of its corresponding random networks at a significance level. For a ncRNA-related ceRNA network or a ncRNA synergistic competition network, if the $p$-value of the Student's $t$-test is smaller than a cutoff (e.g. 0.05), the characteristic path length (or density) of a ncRNA-related ceRNA network or a ncRNA synergistic competition network is significantly smaller (or higher) than that of its corresponding random networks, then the network is a small-world network.

## Network and hub similarity calculation

In terms of ncRNA synergistic competition networks or hub ncRNAs, the network or hub similarity between tumor $c$ and tumor $d$ is computed as follows.

$$sim_{cd} = \frac{|SCR_c \cap SCR_d|}{min(|SCR_c|, |SCR_d|)} \tag{6}$$

where $SCR_c$ and $SCR_d$ denote the identified ncRNA synergistic competition regulation (networks or hub ncRNAs) in tumor $c$ and tumor $d$, respectively, $|SCR_c \cap SCR_d|$ is the number of common ncRNA synergistic competition relationships or hub ncRNAs between $SCR_c$ and $SCR_d$, and $min(|SCR_c|, |SCR_d|)$ denotes the smaller number of ncRNA synergistic competition relationships or hub ncRNAs between $SCR_c$ and $SCR_d$.

## Hub ncRNA identification

Hub ncRNAs are defined as highly connected nodes in a ncRNA synergistic competition network. Here, we use the cumulative probability of Poisson distribution to decide whether a ncRNA is a hub ncRNA:

$$p(x \geq k) = 1 - \sum_{i=0}^{k-1} \frac{\lambda^i e^{-\lambda}}{i!} \tag{7}$$

where $\lambda = np$, $p = \frac{m}{A_n^2}$, $n$ is the number of ncRNAs, $m$ is the number of ncRNA synergistic competition relationships in a ncRNA synergistic competition network, and $A_n^2$ is the number of all possible ncRNA synergistic competition pairs. Smaller $p$-value of a ncRNA indicates that the ncRNA is more likely to be a hub ncRNA. In this work, the cutoff of the $p$-value is set to 0.05.

## Uncovering conserved and rewired interactions and hubs

The ncRNA synergistic competition interaction or hub ncRNA existing in at least two malignant tumors is defined as a conserved ncRNA synergistic competition interaction or hub ncRNA, whereas the ncRNA synergistic competition interaction or hub ncRNA existing in only one malignant tumor is defined as a rewired ncRNA synergistic competition interaction or hub ncRNA. In the case of the identified ncRNA synergistic competition networks and hub ncRNAs across 31 malignant tumors, we can uncover conserved and rewired ncRNA synergistic competition interactions and hub ncRNAs.

## Multi-class classification analysis

To understand the classification performance of the discovered conserved ncRNA synergistic competition interactions and hub ncRNAs, we train a classification model to evaluate the discriminating ability of the features in classifying tumor types, and also train a model to classify BRCA subtypes (LumA, LumB, Her2, Basal and Normal) to understand the ability of the features in discriminating cancer subtypes. We carry out multi-class classification analysis by using the Binary Relevance (BR) [43] strategy in the R package utiml [44] to perform multi-class classification analysis. In the BR strategy, the Support Vector Machine (SVM) [45] implemented in the R package e1071 [46] is selected as the base classifier to create the multi-class model. We use 22 metrics (accuracy, average-precision, clp, coverage, F1, hamming-loss, macro-AUC, macro-F1, macro-precision, macro-recall, margin-loss, micro-AUC, micro-F1, micro-precision, micro-recall, mlp, one-error, precision, ranking-loss, recall, subset-accuracy and wlp as described in **S1 File**) and conduct 10-fold cross-validation to evaluate the

performance of ncRNAs in conserved ncRNA synergistic competitions and hub ncRNAs. Also, we apply a baseline method [47] and a random prediction method in the R package utiml [44] for comparison in the evaluation of the classification performance.

### Survival analysis

To compute the risk score of each malignant tumor sample, the multivariate Cox model [48] in the R package survival [49] is fitted by using ncRNAs in conserved ncRNA synergistic competition and hub ncRNAs as features. According to the risk scores of each malignant tumor sample, the top one-third of the malignant tumor samples are put in the high risk group and the remaining two thirds are put in the low risk group. In terms of the gene expression levels of the ncRNAs, the Log-rank test is applied to assess the distinction of the feature ncRNAs between the high and the low risk groups or between different molecular subtypes. In addition, the proportional hazard ratio (HR) between the high and the low risk groups or between different molecular subtypes is also computed.

### Enrichment analysis

To understand the biological significance of hub ncRNAs across the 31 malignant tumors, we conduct enrichment analysis in three categories, disease, cell marker and cancer hallmark using LncSEA [50]. The hub ncRNAs are enriched in significant disease, cell marker and cancer hallmark terms with an adjusted *p*-value (adjusted by Benjamini-Hochberg method) cutoff of 0.05. Moreover, LncSEA is also used to uncover whether the synergistic competition ncRNAs are involved in drug resistance or not. The synergistic competition ncRNAs are significantly enriched in drug terms with an adjusted *p*-value (adjusted by Benjamini-Hochberg method) less than 0.05.

### Molecular subtyping of malignant tumors

Within the same type of malignant tumor, malignant tumors have great phenotypic differences and genetic heterogeneity. Because of this, molecular subtyping is usually applied to divide malignant tumors into subtypes with distinct molecular and clinical characteristics. To assess the performance of the synergistic competition ncRNAs in molecular subtyping, we use an ensemble method implemented in the CancerSubtypes [51] R package (combining the Similarity Network Fusion (SNF) [52] and Consensus Clustering (CC) [53] methods) to generate malignant tumor subtypes. For each malignant tumor, the number of molecular subtypes is empirically set according to the previous studies on molecular subtyping. (**S3 Data**).

### Immune regulation analysis

Numerous ncRNAs have been shown to play important roles in immune regulation. For the immune regulation analysis, we obtain the significant immune-related ncRNA-pathway and ncRNA-cell pairs from ImmReg [54]. Based on the significance of the enrichment of ncRNAs with immune pathways and the significance of the correlation between ncRNAs and immune cells, we further determine which synergistic competition ncRNAs are significantly enriched in immune pathways and correlated with immune cells in each malignant tumor.

## Results

### The ceRNA regulation landscape in Pan-cancer

Using the transcriptomics data in TCGA, we apply SCOM for detecting two types of ceRNA regulation landscape across the 31 malignant tumors. In the ceRNA regulation landscape, the

two types of ceRNA networks (lncRNA- and pseudogene-related ceRNA networks) have different numbers of ceRNA interactions across the 31 malignant tumors. Generally, the number of ceRNA interactions of the pseudogene-related ceRNA networks is significantly larger than that of the lncRNA-related ceRNA networks (**Fig 2A**, *p*-value = 1.84E-13 with paired Student's *t*-test). This result may be explained that the number of pseudogenes is larger than that of lncRNAs in the transcriptomics data. Network topological analysis reveals that 5 out of the 31 (16.13%) lncRNA-related ceRNA networks and 16 out of the 31 (51.61%) pseudogene-related ceRNA networks follow a power law distribution (**Fig 2B**). After merging the lncRNA and pseudogene related ceRNA networks of each tumor, 14 out of the 31 (45.16%) merged ncRNA-related ceRNA networks follow a power law distribution (**Fig A** in **S1 File**). Moreover, 7 out of the 31 (22.58%) lncRNA-related ceRNA networks and none of the pseudogene-related ceRNA networks have shorter characteristic path lengths than their corresponding random networks, and 30 out of the 31 (96.77%) lncRNA (and pseudogene related) ceRNA networks have higher densities than their corresponding random networks (**Fig 2B**). In the merged ncRNA-related ceRNA networks of each tumor, we have found that none of the ncRNA-related ceRNA networks have shorter characteristic path lengths than their corresponding random networks, and all of the ncRNA-related ceRNA networks have higher densities than their corresponding random networks (**Fig A** in **S1 File**). These observations indicate that less than or nearly half of the ceRNA networks across the 31 malignant tumors tend to be scale-free networks, and most or all of the ceRNA networks across the 31 malignant tumors are likely to be small-world networks with higher densities than their corresponding random networks.

With the lncRNA and pseudogene related ceRNA networks, the similarity intervals between the 31 malignant tumors are [0.13 0.52] and [0.10 0.44], respectively (**Fig 2C**). Moreover, in terms of the ncRNA (including both lncRNA and pseudogene) related ceRNA networks, the similarity interval between the 31 malignant tumors is [0.12 0.47] (**Fig B** in **S1 File**). This result shows that the malignant tumors may have common ncRNA-related ceRNA regulation. We have also discovered that each tumor ceRNA network varies in similarity to other tumor ceRNA networks, revealing known and hidden relationships among these malignant tumors. In addition, we have observed that ~16.45% lncRNA-related ceRNA regulations and ~28.58% pseudogene-related ceRNA regulations only exist in one malignant tumor (**Fig 2D**). Overall, ~24.80% ncRNA-related ceRNA regulations are occurred in one malignant tumor (**Fig C** in **S1 File**). For malignant tumors, the percentages of the rewired ncRNA-related ceRNA regulations are different, demonstrating different degrees of heterogeneity across malignant tumors (**Fig D** in **S1 File**). Particularly, 106 (less than 0.01%) ncRNA-related ceRNA regulations are conserved in the 31 malignant tumors (**Fig 2E**), indicating a low conservation of ceRNA regulation in Pan-cancer. By conducting the multi-class classification and survival analysis, we have shown that the conserved ncRNA-related ceRNA regulations in the 31 malignant tumors have better performance than the baseline method and the random prediction method in classifying 31 malignant tumors, and can distinguish the high and the low risk groups of each malignant tumor at a significant level (Log-rank *p*-value < 0.05 and HR > 2) (**Fig E** in **S1 File**). This result reveals that the conserved ncRNA-related ceRNA regulations may act as a common core of ceRNA regulation across malignant tumors.

## The ncRNA synergistic competition landscape in Pan-cancer

Taking advantage of the transcriptomics data in TCGA and the predicted ncRNA-related ceRNA interactions, we utilise SCOM for uncovering ncRNA (including lncRNAs and pseudogenes) synergistic competition landscape across 31 malignant tumors. Similar to the number of ceRNA interactions, the number of ncRNA synergistic competition interactions

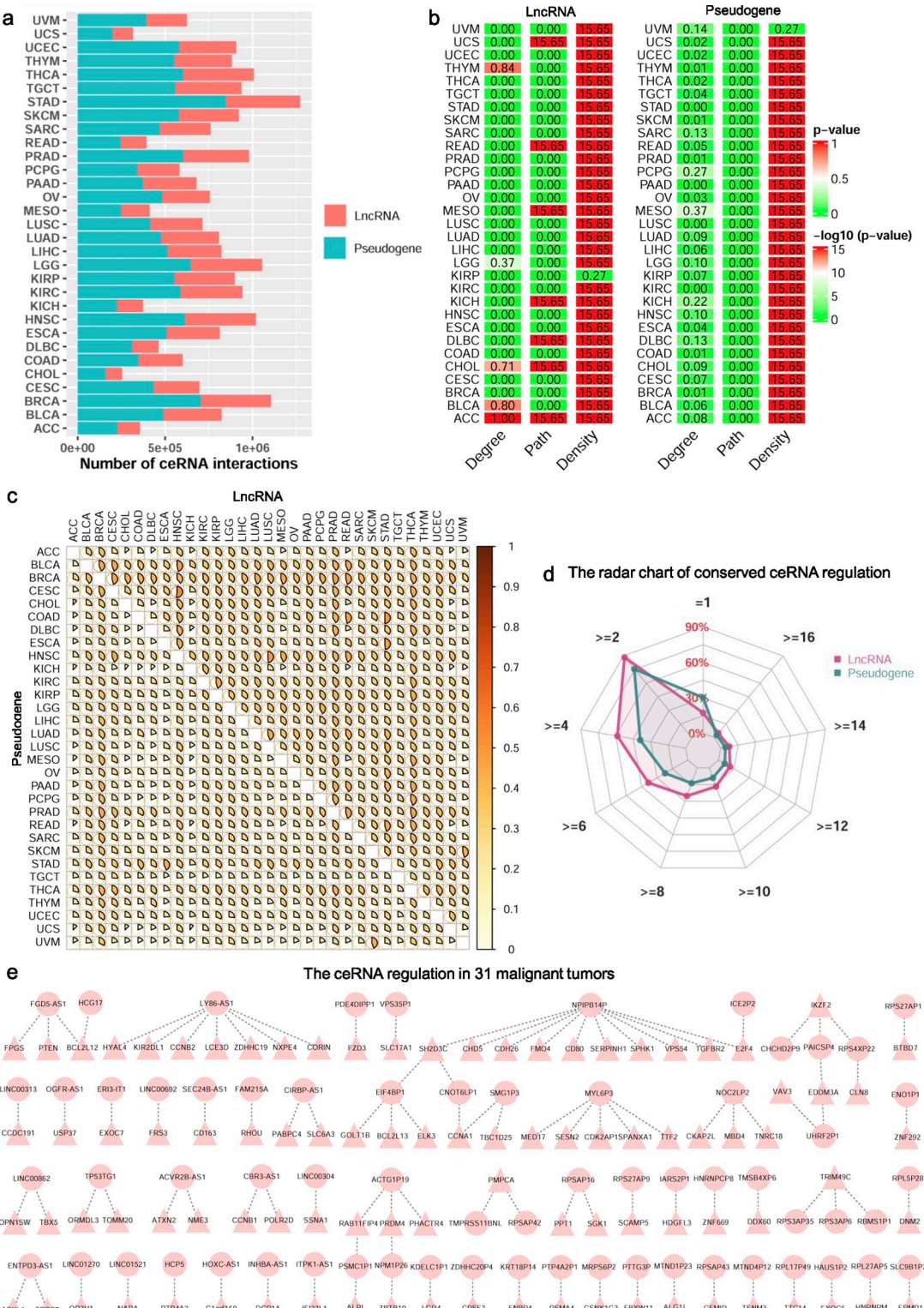

**Fig 2. The ceRNA regulation landscape across malignant tumors.** (a) The number of ncRNA-related ceRNA interactions in each malignant tumor. (b) The topological characteristics of ncRNA-related ceRNA networks for the 31 malignant tumors. The Degree column is the *p*-value of fitting power-law distribution. The *p*-values equal to or more than 0.05 indicate that the degree of ncRNA-related ceRNA networks obeys the power-law distribution. The Path and Density columns represent the –log10(*p*-value) of Student's *t*-test, and the *p*-values less than 0.05 display that the characteristic path lengths (or densities) of ncRNA-

related ceRNA networks are significantly shorter (or higher) than those of their corresponding random networks. (c) The similarity matrix shows the similarity between each pair of lncRNA (upper triangle) and pseudogene (lower triangle) related ceRNA networks across 31 malignant tumors. (d) The radar chart shows the percentage of ceRNA regulations predicted in different numbers of malignant tumors. (e) The conserved ceRNA regulation predicted in 31 malignant tumors.

(including lncRNA-lncRNA, lncRNA-pseudogene and pseudogene-pseudogene synergistic competition pairs) also varies across the 31 malignant tumors. The largest and least numbers of ncRNA synergistic competition interactions are inferred in Stomach adenocarcinoma (STAD) and Cholangiocarcinoma (CHOL), respectively (**Fig 3A**). This result may be explained in part by the different number of ceRNA interactions predicted in the 31 malignant tumors. Network topological analysis shows that 27 out of the 31 (87.10%) ncRNA synergistic competition networks follow a power law distribution (**Fig 3B**). Furthermore, 19 out of the 31 (61.29%) ncRNA synergistic competition networks have shorter characteristic path lengths than their corresponding random networks, and all of the ncRNA synergistic competition networks have higher densities than their corresponding random networks (**Fig 3B**). These findings indicate that most of the ncRNA synergistic competition networks across the 31 malignant tumors are scale-free networks, and small-world networks with shorter characteristic path lengths and higher densities than their corresponding random networks.

With the ncRNA synergistic competition networks, the similarity among the 31 malignant tumors is less than 6% (**Fig 3C**). This low similarity indicates that the malignant tumors may share a small portion of ncRNA synergistic competition with each other. In addition, ~94.06% ncRNA synergistic competition interactions only occur in one malignant tumor, and none of the ncRNA synergistic competition interactions exist in more than six malignant tumors (**Fig 3D**). Specifically, we have observed that the percentage of the conserved ncRNA synergistic competition interactions in at least two malignant tumors is ~5.94%. This result reveals that the ncRNA synergistic competition interactions of different malignant tumors tend to be rewired in their respective tumor microenvironments. To understand whether the ncRNAs of the conserved synergistic competition in at least two malignant tumors are biologically meaningful, we further perform enrichment analysis in three categories including diseases, cell markers and cancer hallmarks. The functional enrichment analysis has revealed that the ncRNAs of the conserved synergistic competition in at least two malignant tumors are significantly enriched in diseases (e.g. Cancer), cell markers (e.g. Exhausted CD8+ T cell), and cancer hallmarks (e.g. Epithelial-mesenchymal transition) common to multiple malignant tumors (**S4 Data**). Moreover, the percentage of the rewired ncRNA synergistic competition for each malignant tumor is all more than 80%, demonstrating a great heterogeneity across malignant tumors in terms of ncRNA synergistic competition (**Fig 3E**).

Based on the topological features of the ncRNA synergistic competition networks, we further identify hub ncRNAs associated with malignant tumors. We have discovered that the number of hub ncRNAs varies across different malignant tumors (**Fig 4A**). In terms of the hub ncRNAs, the similarity between the 31 malignant tumors is [0.07 0.79] (**Fig F** in **S1 File**). This result shows that the malignant tumors may have common hub ncRNAs. In addition, we have observed that the hub ncRNAs can significantly distinguish the high risk and the low risk groups of each malignant tumor (**Fig Ga** in **S1 File**), and the hub ncRNAs covering 96.77% malignant tumors are significantly enriched at least one of Disease, Cell markers and Cancer hallmarks terms (**Fig Gb** in **S1 File** and **S5 Data**). Different from the ncRNA synergistic competition networks, less than 13% hub ncRNAs are only in one malignant tumor but nearly 90% hub ncRNAs exist in at least two malignant tumors (**Fig 4B**). In particular, ~7.63% hub ncRNAs are even conserved in at least 16 malignant tumors. This result reveals that the hub

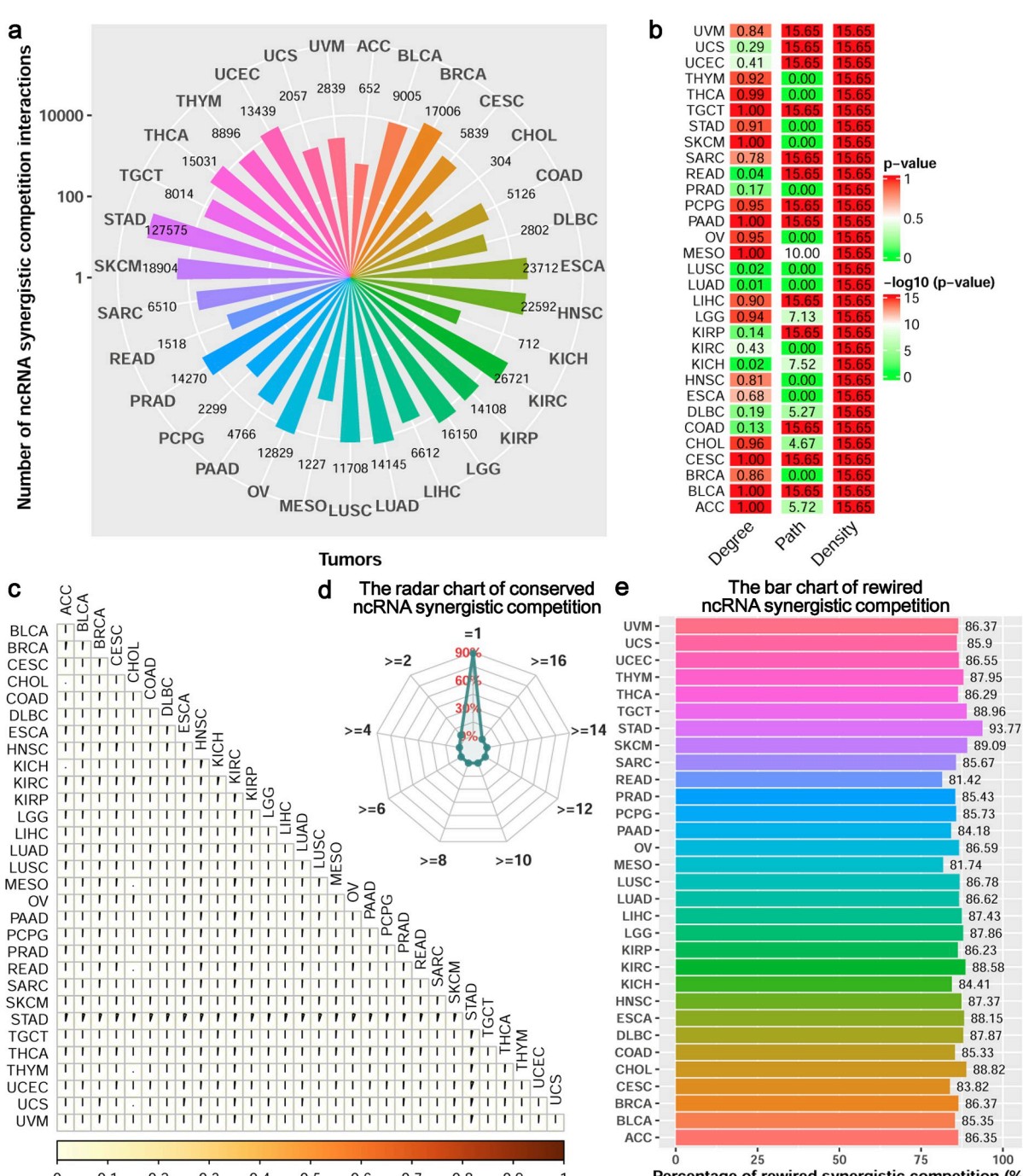

**Fig 3. Network analysis of ncRNA synergistic competition across malignant tumors.** (a) The number of ncRNA synergistic competition interactions in each malignant tumor. (b) The topological characteristics of ncRNA synergistic competition networks for the 31 malignant tumors. The Degree column is the *p*-value of fitting power-law distribution. The *p*-values equal to or more than 0.05 indicate that the degree of the ncRNA synergistic competition networks obeys the power-law distribution. The Path and Density columns represent the–log10(*p*-value) of Student's *t*-test, and the *p*-values less than 0.05 suggest that the characteristic path lengths (or densities) of ncRNA synergistic competition networks are significantly shorter (or higher) than those of their corresponding random networks. (c) The similarity matrix showing the similarity between each pair of ncRNA synergistic competition networks across 31 malignant tumors. (d) The radar chart showing the percentages of ncRNA synergistic competition interactions predicted in different number of malignant tumors. (e) The bar chart showing the percentages of ncRNA synergistic competition interactions predicted in one malignant tumor.

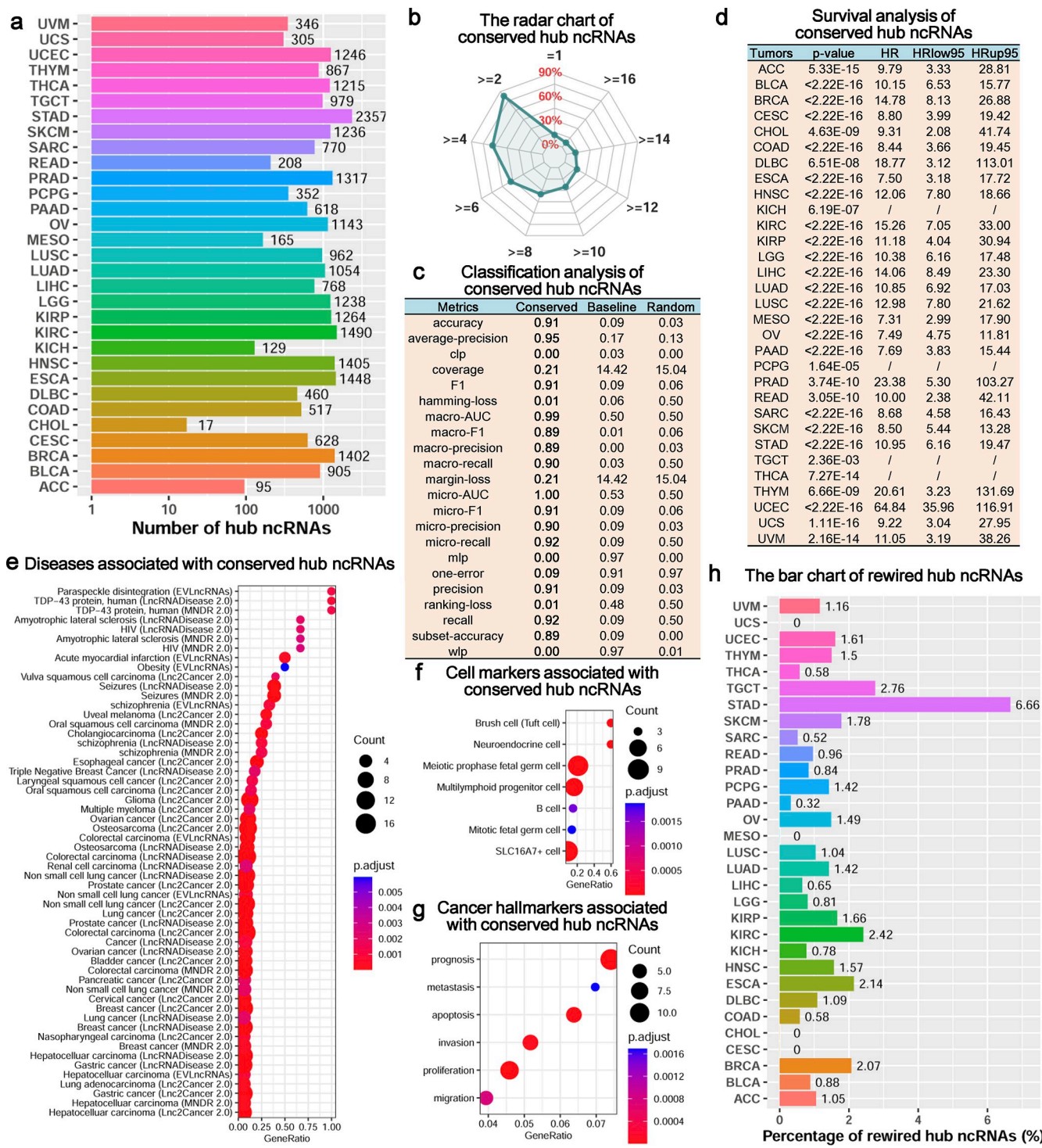

**Fig 4. Hub analysis of ncRNA synergistic competition across malignant tumors.** (a) The number of hub ncRNAs in each malignant tumor. (b) The radar chart displaying the percentage of hub ncRNAs predicted in different number of malignant tumors. (c) Multi-class classification analysis of conserved hub ncRNAs in at least 16 malignant tumors. (d) Survival analysis of conserved hub ncRNAs in at least 16 malignant tumors. The symbol "/" stands for an infinite value. (e-g) Enrichment analysis (three categories including disease, cell marker and cancer hallmark) of conserved hub ncRNAs in at least 16 malignant tumors. (h) The bar chart indicating the percentage of hub ncRNAs predicted in one malignant tumor.

ncRNAs of different malignant tumors tend to be conserved in their respective tumor microenvironments.

To understand whether there is a common core of hub ncRNAs across malignant tumors, we investigate the conserved hub ncRNAs occurred in at least 16 malignant tumors. Functional analysis indicates that the conserved hub ncRNAs have better performance than the baseline method and the random prediction method in classifying the 31 malignant tumors (**Fig 4C**), and can significantly distinguish the high risk and the low risk groups of each malignant tumor (**Fig 4D**). This result shows that the conserved hub ncRNAs may act as potential diagnostic and prognostic biomarkers across malignant tumors. Moreover, the conserved hub ncRNAs are significantly enriched in diseases (**Fig 4E**), cell markers (**Fig 4F**) and cancer hallmarks (**Fig 4G**) common to several malignant tumors. This result suggests that the malignant tumors may have common cell markers and cancer hallmarks.

In addition to the conserved hub ncRNAs, the rewired hub ncRNAs switching from tumor to tumor are important to understand the heterogeneity of malignant tumors. We have found that the percentage of the rewired hub ncRNAs for each malignant tumor is less than 7%, revealing a small heterogeneity across malignant tumors in terms of hub ncRNAs (**Fig 4H**). Multi-class classification analysis shows that the rewired hub ncRNAs have better performance than the baseline method and the random prediction method in classifying the 31 malignant tumors (**Fig Gc** in **S1 File**). Moreover, survival analysis displays that the rewired hub ncRNAs can significantly distinguish the high risk and the low risk groups of each malignant tumor (**Fig Gd** in **S1 File**). This result suggests that the rewired hub ncRNAs may also act as potential diagnostic and prognostic biomarkers across malignant tumors.

## Synergistic competition ncRNAs are involved in drug resistance

It is reported that ncRNAs play important roles in drug resistance, and the dysregulation of them can induce resistance to anti-cancer therapy and cause a failure of malignant tumor clinical treatment [55,56]. Drug enrichment analysis shows that the ncRNA synergistic competition networks of ~96.77% of the 31 malignant tumors and the hub ncRNAs of ~64.52% of the 31 malignant tumors are significantly enriched in at least one drug resistance term (**Fig 5A** and **S7 and S6 Data**). We have found that the ncRNA synergistic competition networks of 29 malignant tumors are involved in the drug *cisplatin* resistance (**Fig 5B**). The drug *cisplatin* is a well-known chemotherapy medication, and can be used to treat a number of cancers including bladder cancer, brain tumors, cervical cancer, esophageal cancer, head and neck cancer, lung cancer, mesothelioma, ovarian cancer, etc [57]. Moreover, the resistance of two drugs *L-685458* and *LBW242* is closely associated with the ncRNA synergistic competition networks of 4 malignant tumors (**Fig 5B**). The drug *L-685458* is a potent transition state analog (TSA) γ-secretase inhibitor (GSI) and is an attractive therapeutic candidate for Alzheimer Disease (AD) and cancers [58], and the drug *LBW242* is a potent and orally active proapoptotic IAP inhibitor and potentiates treat multiple myeloma and ovarian cancer [59,60]. Similar to the ncRNA synergistic competition networks, the top three enriched drugs of hub ncRNAs across malignant tumors are also *cisplatin*, *L-685458* and *LBW242* (**Fig 5C**). This result suggests that the synergistic competition ncRNAs (i.e. ncRNAs involved in the synergistic competition) across malignant tumors are involved in drug resistance, and may act as targets for anti-cancer therapy.

## Synergistic competition ncRNAs contribute to distinguishing molecular subtypes of malignant tumors

Malignant tumors are a collection of heterogeneous diseases with both phenotypic and genetic heterogeneity, and a malignant tumor usually possess multiple distinct molecular subtypes

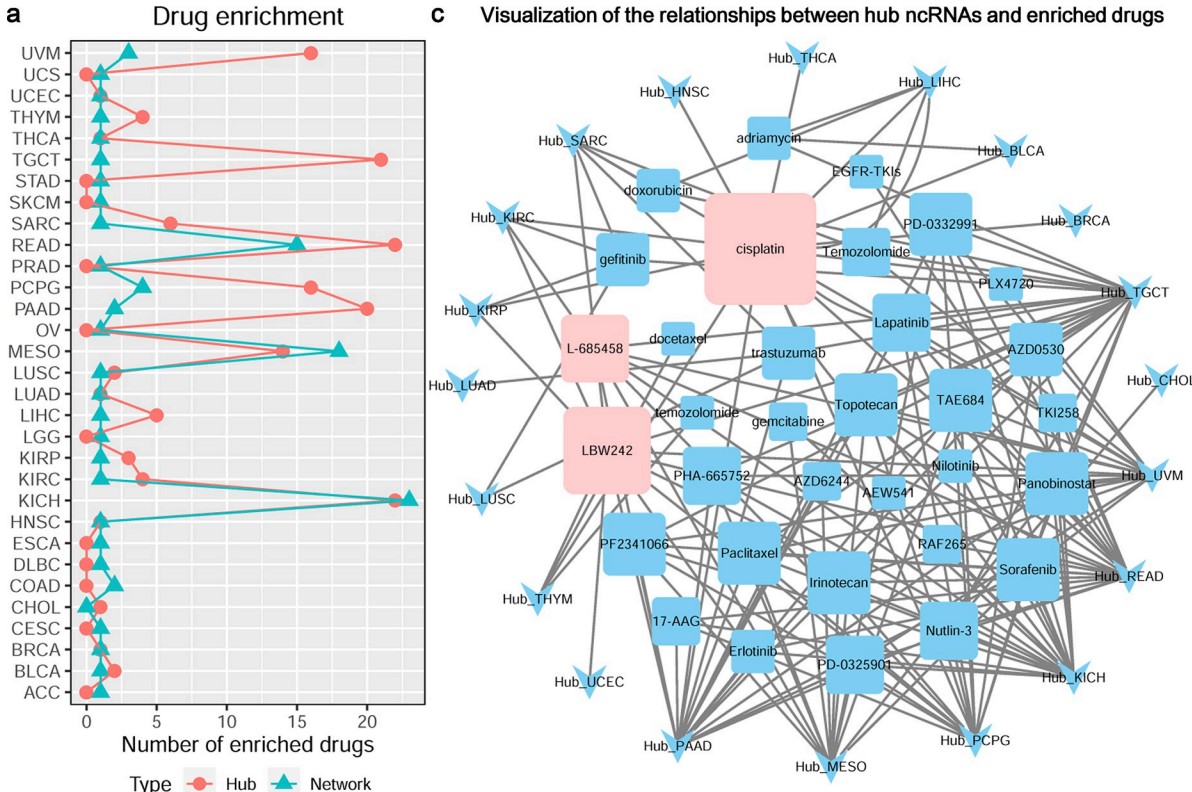

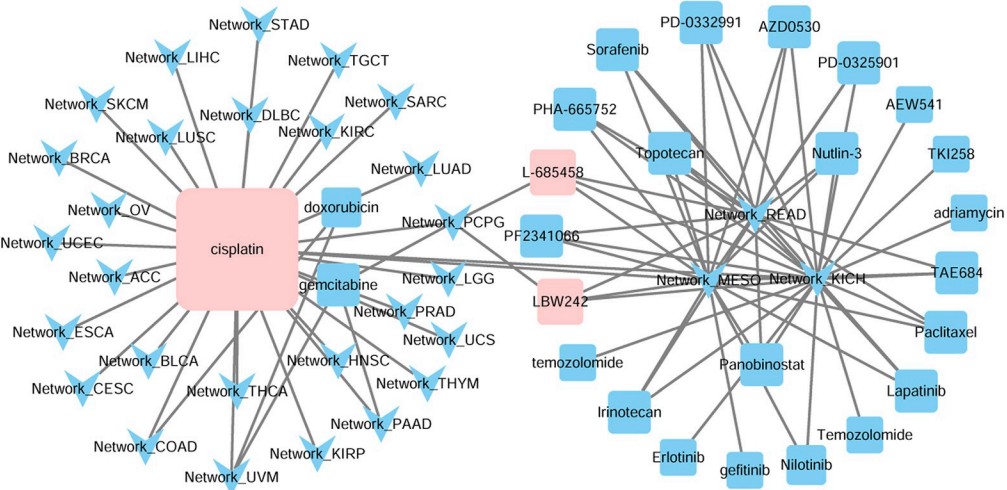

**Fig 5. Drug resistance analysis of ncRNA synergistic competition networks and hub ncRNAs across malignant tumors.** (a) The number of significantly enriched drugs for each malignant tumor. (b) Relationships between ncRNA synergistic competition networks and enriched drugs. For example, the node "Network_ACC" denotes the ncRNA synergistic competition network in ACC. Larger drug nodes denote more links with ncRNA synergistic competition networks in malignant cancers. (c) Relationships between hub ncRNAs and enriched drugs. For example, the node "Hub_ACC" denotes the hub ncRNAs in ACC. Larger drug nodes denote more links with hub ncRNAs in malignant cancers.

[61]. Molecular subtyping analysis of the synergistic competition ncRNAs (i.e. ncRNAs involved in the synergistic competition) shows that the average silhouette width of the molecular subtyping of ~90.32% of the 31 malignant tumors is equal to or greater than 0.60 (**Fig 6A**),

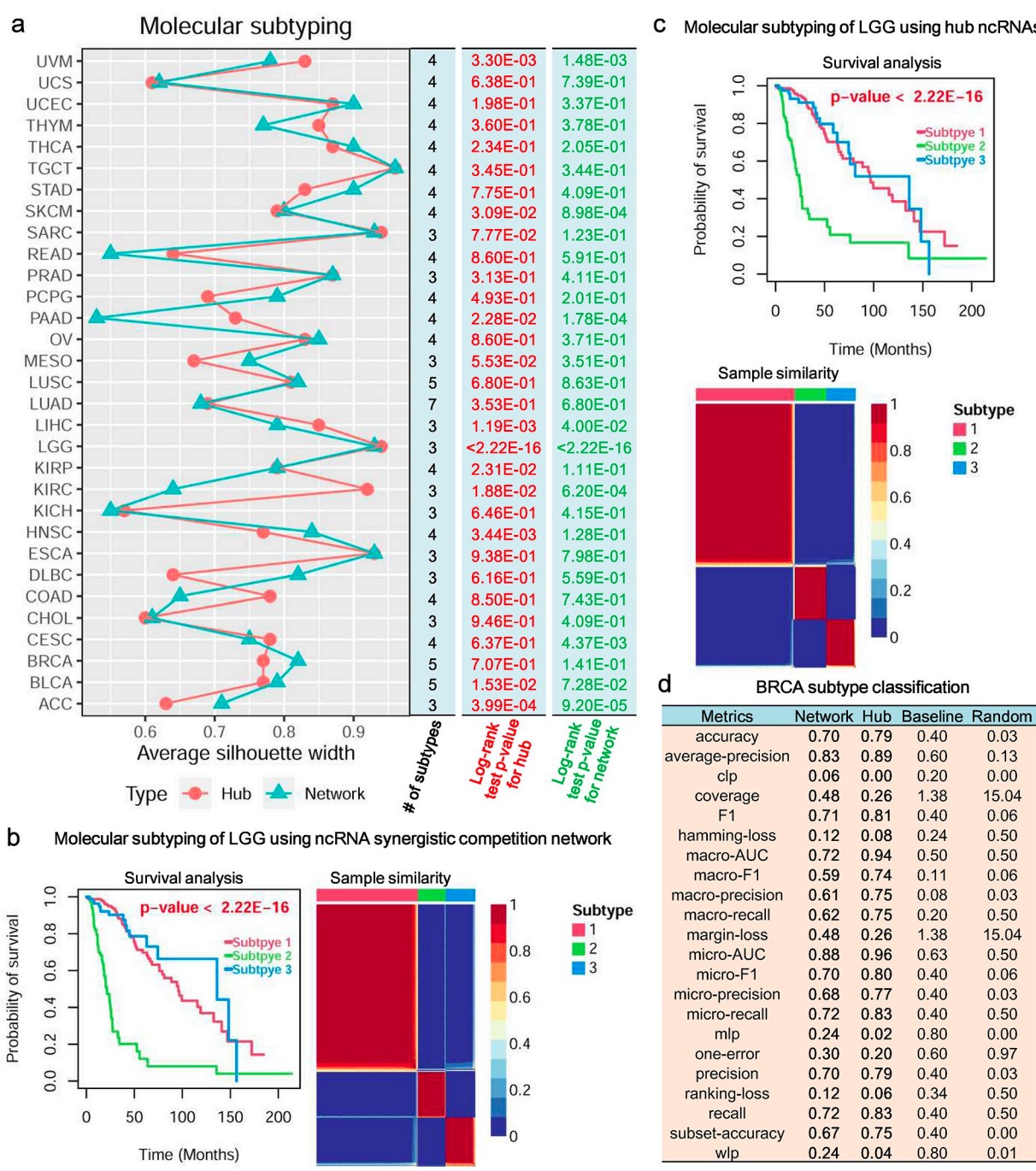

**Fig 6. Molecular subtype analysis of malignant tumors.** (a) Molecular subtyping of the 31 malignant tumors using ncRNA synergistic competition networks and hub ncRNAs. Higher average silhouette width indicates that a malignant tumor sample is better matched with the identified molecular subtypes. (b) Molecular subtyping of LGG using ncRNA synergistic competition network. (c) Molecular subtyping of LGG using hub ncRNAs. (d) Classification analysis of BRCA subtypes.

indicating that the synergistic competition ncRNAs can well assign malignant tumor samples into distinct molecular subtypes. In the case of using hub ncRNAs, the molecular subtypes of ~32.26% of the 31 malignant tumors can be significantly distinguished (Log-rank test *p*-value < 0.05, **Fig 6A**). Moreover, in the case of using ncRNA synergistic competition

networks, the molecular subtypes of ~25.81% of the 31 malignant tumors can be significantly distinguished (Log-rank test *p*-value < 0.05, **Fig 6A**). For seven malignant tumor types, including Adrenocortical carcinoma (ACC), Kidney renal clear cell carcinoma (KIRC), Brain Lower Grade Glioma (LGG), Liver hepatocellular carcinoma (LIHC), Pancreatic adenocarcinoma (PAAD), Skin Cutaneous Melanoma (SKCM) and Uveal Melanoma (UVM), the hub ncRNAs and ncRNA synergistic competition networks all have contributions to distinguish molecular subtypes of them. For example, in the case of using hub ncRNAs and ncRNA synergistic competition networks, LGG can be divided into three distinct molecular subtypes (**Fig 6B** and **6C**). This result reveals that the synergistic competition ncRNAs contribute to distinguishing molecular subtypes of malignant tumors, and may act as prognostic biomarkers in several malignant tumors.

Since the molecular subtypes (including Luminal A, Luminal B, HER-2, Basal, Normal-like) of Breast invasive carcinoma (BRCA) are mostly accepted by researchers, we focus on studying the BRCA subtype classification performance by using hub ncRNAs and ncRNA synergistic competition networks in BRCA. By applying multi-class classification analysis into the BRCA datasets from TCGA and Molecular Taxonomy of Breast Cancer International Consortium (METABRIC) [62], we have shown that the synergistic competition ncRNAs have better performance than the baseline method and the random prediction method in classifying BRCA subtypes (**Fig 6D** and **Table A** in **S1 File**), suggesting that the synergistic competition ncRNAs may act as potential diagnostic biomarkers of BRCA.

## Synergistic competition ncRNAs participate in immune regulation

As regulatory elements in immune system, ncRNAs can influence immune development and function, leading to immune diseases including autoimmunity and haematological cancers [63,64]. Immune regulation analysis reveals that the ncRNA synergistic competition networks of all the 31 malignant tumors and the hub ncRNAs of 30 malignant tumors are significantly involved in at least one immune-related pathway (**Fig 7A** and **S8** and **S9 Data**). For instance, the ncRNA synergistic competition networks of the 31 malignant tumors are closely associated with six immune-related pathways (*Antigen Processing and Presentation*, *Cytokine Receptors*, *Antimicrobials*, *Cytokines*, *Natural Killer Cell Cytotoxicity*, *TCR signaling Pathway*) (**Fig 7C**). In addition, the hub ncRNAs of 29 malignant tumors are closely related to two immune-related pathways (*Antigen Processing and Presentation*, *Cytokine Receptors*) (**Fig 7D**). Moreover, the ncRNA synergistic competition networks and the hub ncRNAs of 31 malignant tumors are significantly associated with at least 40 immune cells (**Fig 7B** and **S10** and **S11 Data**). This result suggests that the synergistic competition ncRNAs (i.e. ncRNAs involved in the synergistic competition) take part in immune regulation, and the dysregulation of them may influence the occurrence and development of immune diseases including malignant tumors.

## SCOMdb: a web-based resource for ncRNA synergistic competition in Pan-cancer

To provide a convenient and web-accessible resource of ncRNA synergistic competition landscape across malignant tumors for biologists, we have developed a comprehensive and interactive database, SCOMdb (https://comblab.cn/SCOMdb/). By using this web-based resource, users can browse the ceRNA interactions in specific malignant tumor of interest (**Fig 8A**). It also provides the ncRNA synergistic competition interactions in malignant tumors (**Fig 8B**). In addition, users can query the hub ncRNAs of interest across malignant tumors (**Fig 8C**). All the data in this work can be downloaded for downstream analysis (**Fig 8D**). To help users

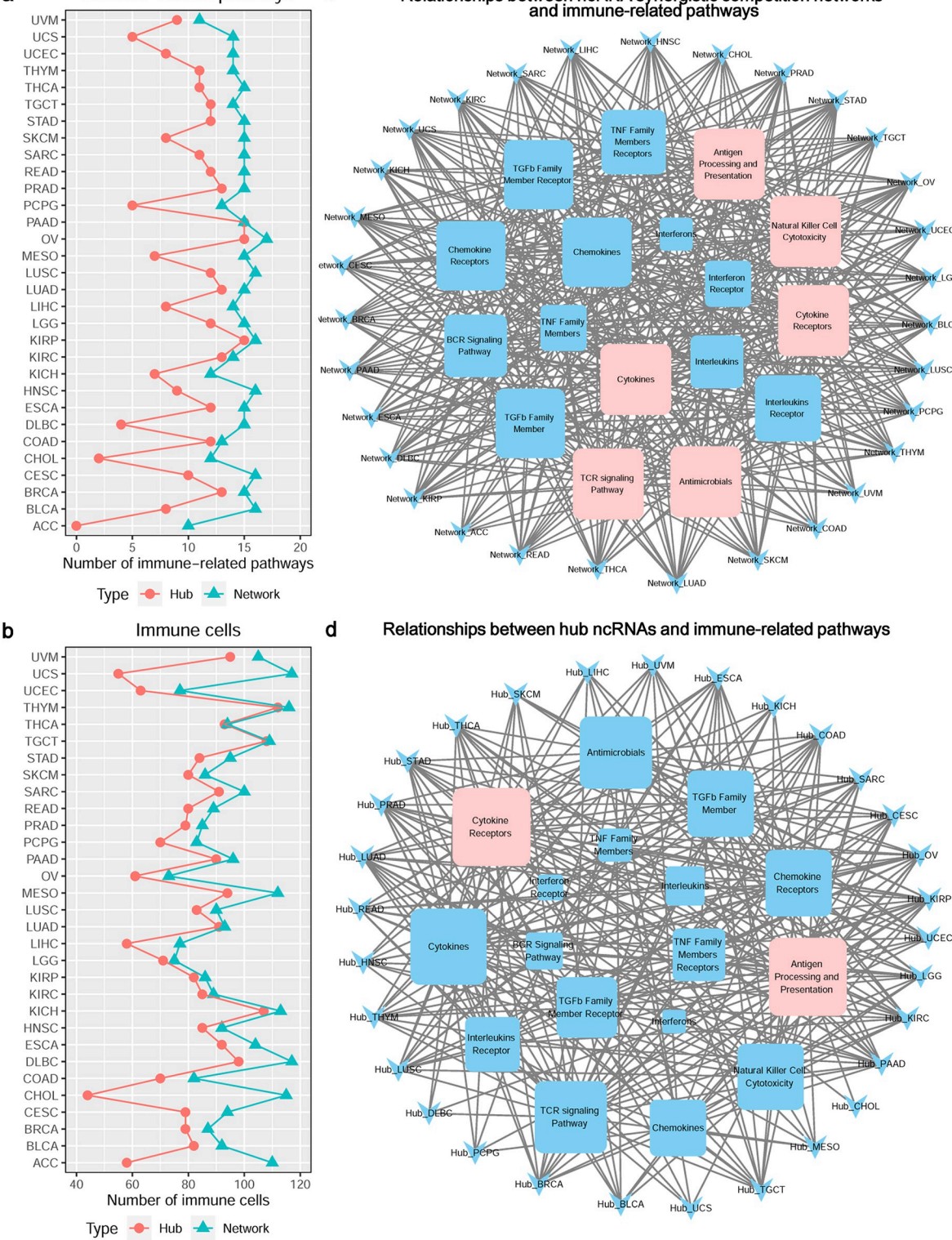

**Fig 7. Immune regulation analysis of synergistic competition ncRNAs across malignant tumors.** (a) The number of significant immune-related pathways associated with each malignant tumor. (b) The number of significant immune cells associated with each malignant tumor. (c) Relationships between ncRNA synergistic competition networks and immune-related pathways. For example, the node "Network_ACC" denotes the ncRNA synergistic competition network in ACC. Larger immune-related pathway nodes denote more links with ncRNA synergistic competition networks in malignant cancers. (d) Relationships between hub ncRNAs and immune-related

pathways. For example, the node "Hub_ACC" denotes the hub ncRNAs in ACC. Larger immune-related pathway nodes denote more links with hub ncRNAs in malignant cancers.

apply SCOM to new datasets, we have also provided the SCOM R package for inferring ncRNA synergistic competition by integrating sample-matched gene (ncRNA and mRNA) expression data and putative miRNA-target interactions (**Fig 8E**). Different from the existing databases of collecting ceRNA interactions (e.g. ENCORI [33], SPONGEdb [65], LncACTdb [66], and LnCeCell [67], see [68] for a summary of databases for ceRNAs), SCOMdb concentrates on providing the synergistic competition interactions of ncRNAs acting as ceRNAs. The provided resources of SCOM will be continuously updated, and can serve as a valuable resource for biologists interested in investigating ncRNA synergistic competition across malignant tumors.

## Conclusions and discussion

Abundant evidence reveals that ncRNAs can serve as important regulators across malignant tumors. However, from the perspective of synergistic competition, ncRNA regulation is still unearthed. In this work, we present the *ncRNA synergistic competition hypothesis* and the SCOM framework, and further elaborate the application of the SCOM framework to comprehensively infer the synergistic competition ncRNAs that potentially act as carcinogenic biomarkers. From the perspective of synergistic competition, we have shown that the synergistic competition ncRNAs are potential diagnostic and prognostic biomarkers of malignant tumors. Moreover, the synergistic competition ncRNAs are likely to involve in drug resistance, contribute to distinguishing molecular subtypes of malignant tumors, and participate in immune regulation. The web-based database SCOMdb (https://comblab.cn/SCOMdb/) provides a valuable resource for exploring ncRNA regulation across malignant tumors.

As important regulators of gene expression, ncRNAs play important roles in the physiology and development of malignant tumors. Yet, the experimentally validated carcinogenic biomarkers are far from complete. Thus, it is challenging to confirm the accuracy of the findings by the SCOM framework. To show the effectiveness of SCOM, we conduct a series of functional analyses including enrichment analysis, molecular subtyping analysis, multi-class classification analysis, survival analysis, and immune regulation analysis for the synergistic competition ncRNAs. We have found that the conserved ncRNA synergistic competition ncRNAs and hub ncRNAs are significantly enriched in diseases, cell markers and cancer hallmarks. In addition, the conserved and rewired hub ncRNAs may act as potential diagnostic and prognostic biomarkers across malignant tumors. Moreover, the synergistic competition ncRNAs are likely to be involved in drug resistance, contribute to distinguishing molecular subtypes of malignant tumors, and participate in immune regulation. These results indicate that the SCOM framework can effectively infer the synergistic competition ncRNAs with biological significance. Although experimentally validated carcinogenic biomarkers are limited, we have discovered that a significant number of synergistic competition ncRNAs are confirmed as carcinogenic biomarkers by the literature (**Fig H** in **S1 File** and **S12 Data**).

The findings in this paper have provided several biological insights although the identified synergistic competition ncRNAs need to be further confirmed by subsequent biological experiments. Firstly, not all the ncRNA synergistic competition networks follow a power law distribution. For example, the ncRNA synergistic competition networks in Kidney Chromophobe (KICH), Lung adenocarcinoma (LUAD), Lung squamous cell carcinoma (LUSC) and Rectum adenocarcinoma (READ) do not obey the power-law distribution. This result indicates that the synergistic competition ncRNAs in these four malignant tumors might equally interact

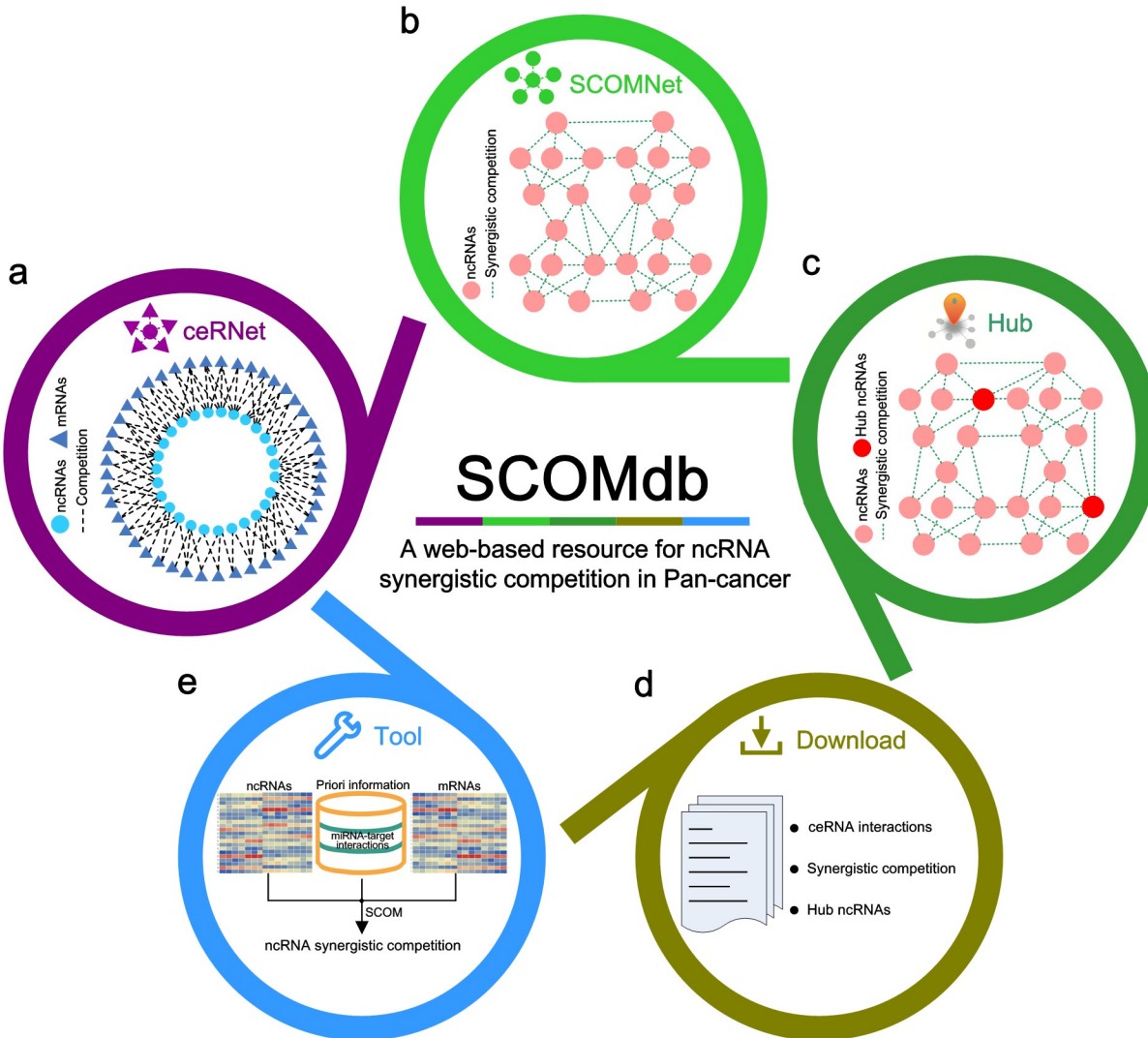

**Fig 8. Diagram of SCOMdb database.** (a) The search of ceRNA regulation in Pan-cancer. (b) The search of ncRNA synergistic competition in Pan-cancer. (c) The search of hub ncRNAs in Pan-cancer. (d) All resources are provided to be downloaded for further analysis. (e) The SCOM method is available for users to study ncRNA synergistic competition by using their own datasets.

with each other. Secondly, a majority of the ncRNA interactions involved in the synergistic competition are rewired, but a minority of hub ncRNAs is rewired. This result suggests that malignant tumors are likely to share hub ncRNAs rather than the ncRNA interactions involved in the synergistic competition. Thirdly, the similarity of hub ncRNAs between the 31 malignant tumors is significantly larger than that of ncRNA synergistic competition networks ($p$-value < 2.20E-16 with paired Student's $t$-test). Moreover, the percentage of rewired ncRNA synergistic competition interactions is significantly larger than that of rewired hub ncRNAs ($p$-value < 2.20E-16 with paired Student's $t$-test). This result reveals that although malignant tumors have great differences in ncRNA synergistic competition networks, the hub ncRNAs between malignant tumors tend to be similar. Fourthly, the synergistic competition ncRNAs can distinguish five BRCA subtypes with accuracy equal to or more than 70%. This result shows that the synergistic competition ncRNAs might be good diagnostic biomarkers in BRCA. Fifthly, we have used in this study the transcriptomics data in TCGA [28], which is one

of the most well-known cancer genomics programs. However, the SCOM framework can also be applied to the transcriptomics data in other sources (e.g. CCLE [69] and MET500 [70]) to investigate ncRNA synergistic competition. Finally, SCOM is not limited to the exploration of lncRNA and pseudogene synergistic competition. It can also be used to infer the synergistic competition of other ncRNAs (e.g. circRNAs) that are important modulators to malignant tumors.

The prediction of ncRNA-related ceRNA network is a critical step to infer ncRNA synergistic competition. Yet, there is still a lack of study on the prediction of ncRNA-related ceRNA networks. In the current work, we apply three criteria (significant sharing of miRNAs, significant positive correlation, and significant sensitive correlation conditioning on shared miRNAs) to predict ncRNA-related ceRNA networks in the 31 malignant tumors. Since the three criteria have been successfully used to investigate ceRNA regulation [27,35,71–73], the ncRNA-related ceRNA networks predicted by SCOM are promising. However, SCOM doesn't consider the stoichiometry of miRNAs and ncRNAs when predicting ceRNA networks. Since miRNA efficacy is determined by its cellular abundance [74] or its abundance within RNA-induced silencing complexes (RISC) [75], the stoichiometry of miRNAs and ncRNAs is considered as a vital factor to predict ceRNA interactions of a ncRNA of interest [76]. An implication of understanding the stoichiometry of miRNAs and ncRNAs is that ncRNAs with higher expression levels are more likely to have greater competitive ability [10]. Therefore, to accurately predict ncRNA-related ceRNA interactions, SCOM will select highly expressed ncRNAs across malignant tumors in future. With deeper understanding of ceRNA mechanisms, we can more accurately predict ncRNA-related ceRNA networks to improve the inference of ncRNA synergistic competition and accelerate the discovery of carcinogenic biomarkers.

For multi-class classification analysis, we use the expression data of ceRNAs as input to train a classification model. Previous studies [77,78] have shown that using module (i.e. a set of ceRNAs) enrichment scores inferred from gene expression data can help to understand the regulatory activity of ceRNA modules, and is a promising way to classify tumor types or subtypes. As a future work, we plan to incorporate the enrichment scores of ceRNA modules with spongEffects [78] to SCOM, and further them as part of the input to train a classification model when classifying tumor types or subtypes.

It is meaningful to design appropriate wet-lab experiments to confirm the predicted ncRNA synergistic competition relationships. For biological experiments, biologists can choose some ncRNAs of interest for validation. For example, to confirm whether *PVT1* and *H19* have synergistic competition relationships or not, biologists can develop an experimental framework to compare the suppressive effects of *PVT1* and *H19* on miRNA activity before and after knocking out one of them. In the two knockout experiments (one for *PVT1* and one for *H19*), if we can observe that several miRNAs show relatively stronger activity, *PVT1* and *H19* have synergistic competition relationships.

In this work, SCOM is focused on studying the synergistic competition between pairs of ncRNAs. However, mRNAs, when acting as ceRNAs, are also important members in ceRNA networks. Therefore, the pairs of mRNAs and the mixed pairs of ncRNAs/mRNAs are also likely to synergistically compete with miRNA targets. In the future, we will extend SCOM for investigating the synergistic competition between the pairs of mRNAs and the mixed pairs of ncRNAs/mRNAs as well.

Here, we focus on studying ncRNA synergistic competition by only taking account of post-transcriptional regulation by miRNAs. However, in addition to miRNAs, transcription factors (TFs) also have a widespread effect on gene expression, and TFs and miRNAs are frequently coordinated to control the expression of their downstream genes [79]. In future, through considering both transcriptional regulation by TFs and post-transcriptional regulation by

miRNAs, it is a promising direction to explore the interplay between TFs and miRNAs in the ceRNA regulation.

Given gene expression data and miRNA-target interactions, SCOM constructs the landscape of ncRNA synergistic competition across malignant tumors at the bulk level. Recently, increasingly advanced single-cell and spatial RNA sequencing technology has opened a way for exploring ncRNA synergistic competition at the single-cell level. With the advance of single-cell and spatial RNA sequencing technology, SCOM will contribute to uncover more carcinogenic biomarkers.

ncRNAs are emerging as potential carcinogenic biomarkers. It is important for us to understand the regulation of ncRNAs in malignant tumors. In-depth study of synergistic competition ncRNAs inferred in this study would contribute to the design of reliable diagnosis and treatment biomarkers for human malignant tumors.

## Supporting information

**S1 Data. The matched tumor types in Pan-cancer dataset.**
(XLSX)

**S2 Data. Clinical information of the matched 8001 tumor samples.**
(XLSX)

**S3 Data. The molecular subtypes of malignant tumors in Pan-cancer dataset.**
(XLSX)

**S4 Data. Enrichment analysis of the conserved ncRNA synergistic competition in at least two malignant tumors.**
(XLSX)

**S5 Data. Enrichment analysis of hub ncRNAs across malignant tumors.**
(XLSX)

**S6 Data. Drug enrichment of ncRNA synergistic competition networks.**
(XLSX)

**S7 Data. Drug enrichment of hub ncRNAs.**
(XLSX)

**S8 Data. Relationships between ncRNA synergistic competition networks and immune-related pathways across malignant tumors.**
(XLSX)

**S9 Data. Relationships between hub ncRNAs and immune-related pathways across malignant tumors.**
(XLSX)

**S10 Data. Relationships between ncRNA synergistic competition networks and immune cells across malignant tumors.**
(XLSX)

**S11 Data. Relationships between hub ncRNAs and immune cells across malignant tumors.**
(XLSX)

**S12 Data. Experimentally validated synergistic competition ncRNAs as biomarkers across malignant tumors.**
(XLSX)

**S1 File. Supporting file.**
(DOCX)

# Author Contributions

**Conceptualization:** Junpeng Zhang, Thuc Duy Le.

**Data curation:** Junpeng Zhang, Thuc Duy Le.

**Formal analysis:** Junpeng Zhang, Xuemei Wei, Chunwen Zhao.

**Funding acquisition:** Junpeng Zhang, Chunwen Zhao, Thuc Duy Le.

**Investigation:** Junpeng Zhang, Lin Liu, Jiuyong Li, Thuc Duy Le.

**Methodology:** Junpeng Zhang, Lin Liu, Xuemei Wei, Chunwen Zhao, Sijing Li, Jiuyong Li.

**Project administration:** Junpeng Zhang, Thuc Duy Le.

**Resources:** Junpeng Zhang, Lin Liu, Jiuyong Li, Thuc Duy Le.

**Software:** Junpeng Zhang.

**Supervision:** Junpeng Zhang, Thuc Duy Le.

**Validation:** Lin Liu, Jiuyong Li.

**Visualization:** Xuemei Wei, Chunwen Zhao, Sijing Li.

**Writing – original draft:** Junpeng Zhang, Lin Liu.

**Writing – review & editing:** Junpeng Zhang, Lin Liu, Jiuyong Li, Thuc Duy Le.

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
