## [Decision Letter · Decision Letter 0]

14 Aug 2023

Dear Dr. Zhang,

Thank you very much for submitting your manuscript "Pan-cancer characterization of ncRNA synergistic competition uncovers potential carcinogenic biomarkers" for consideration at PLOS Computational Biology.

As with all papers reviewed by the journal, your manuscript was reviewed by members of the editorial board and by several independent reviewers. In light of the reviews (below this email), we would like to invite the resubmission of a significantly-revised version that takes into account the reviewers' comments.

We cannot make any decision about publication until we have seen the revised manuscript and your response to the reviewers' comments. Your revised manuscript is also likely to be sent to reviewers for further evaluation.

Sincerely,

Mark Ziemann

Academic Editor

PLOS Computational Biology

Pedro Mendes

Section Editor

PLOS Computational Biology

Reviewer's Responses to Questions

**Comments to the Authors:**

Reviewer #1: Non-coding RNAs (ncRNAs) have emerged as significant regulators of gene expression and have been found to play crucial roles in the physiology and development of malignant tumors. The paper “Pan-cancer characterization of ncRNA synergistic competition uncovers potential carcinogenic biomarkers” provided a framework named as SCOM to show the synergistic competition of competing endogenous RNA (ceRNA), and further present the application of SCOM in drug resistance, molecular subtypes of malignant tumors, and immune regulation. A web-based database SCOMdb has been developed as part of this work, serving as a valuable resource for investigating ncRNA regulation.

The manuscript addressed a relevant question in an emerging field of investigation, however there are several limitations that hamper of the work.

1. The regulation of ncRNA-mRNA interaction needs the presence of transcription factors and miRNAs frequently. The authors should also check for the interaction among miRNA and TFs, and see if they could be involved in a more complex transcriptional circuit (ceRNAs+miRNAs+TFs+TF_targeting mRNAs).

2. We always consider the expressions of ncRNAs and mRNAs, and the interaction networks are tissue or tumor specific. The study is pancancer characteristic. What do the authors think about this issue? How the consistency of molecular subtypes of malignant tumors and types of cancers? What is the application and significancy of this framework in clinical experience?

3. In the method to recruit ncRNA, the criterion requires that ncRNA i and ncRNA j are positively correlated. How about the negative correlated ncRNAs？

4. The authors used the Pan-cancer transcriptomics data including the expression data of ncRNAs (including miRNAs, lncRNAs and pseudogenes) and mRNA. An important point remains to be highlighted that the stoichiometric role of these molecules. This point is referred to as the equimolarity assumption which leads to the prediction of potential ceRNA competition effect around the molecular equilibrium.

Reviewer #2: In their manuscript "Pan-cancer characterization of ncRNA synergistic competition uncovers potential carcinogenic biomarkers", Junpeng Zhang et al. describe a new method SCOM which focuses on unraveling synergistic competition in competing endogenous RNA (ceRNA) networks. Following the hypothesis of ceRNA by Salmena et al., transcripts carrying microRNA binding sites compete for a limited pool of microRNA, affecting each other through de-repression. This effect is investigated here from a new perspective of synergy between ceRNAs, where multiple ceRNAs may boost each other's effect on the network by their joint activity. ceRNAs in synergistic competition are in SCOM those non-coding RNAs that show a significant overlap of mRNA targets (hypergeometric test), have significant positive correlation themselves and both show significant sensitivity correlation considering shared mRNA targets. Notably, sensitivity correlation is used here to first infer the ceRNA network (as it was used before by Paci et al. (2014) and List et al. (2019)) and then again to investigate the competition of two non-coding RNAs conditioning on shared mRNAs. The method is described in sufficient details to allow reproducing it. The results are comprehensive and deeply characterized from multiple angles to help readers understand the meaning of the resulting networks.

# Major

- I welcome the idea to look for synergies between ceRNAs in co-regulating the network. However, by using the concept of sensitivity correlation twice, once for establishing the shared ceRNA network and again, by studying competition of pairs of non-coding RNAs, I wonder if this represents a case of double-dipping as the correlation coefficients considered here are included in both tests. To avoid this, one should split the data and use part to infer the ceRNA network while using the other part to study synergistic competition.

- It is not clear why the authors suspect synergistic competition only between pairs of non-coding RNAs. Pairs of mRNAs and mixed pairs of non-coding / coding RNAs quite likely also exhibit this behavior, especially given that the most powerful ceRNAs in a network are often mRNAs (as shown in the SPONGE paper, List et al. 2019). I can imagine that the authors opted for this split to keep a clear separation between regulatory (non-coding RNAs) and regulated genes (mRNAs) but since all transcripts can in principle fill both roles, this limitation of the current study should be discussed critically.

- In the conclusion, the authors mention the importance of stoichiometry for the effectiveness of ceRNA regulation. A more basic step that I would have expected here is to check for the expression of the synergistic ceRNAs - if those expression values are very low, they are not likely to have a substantial effect; it may thus be better to focus on the more highly expressed examples or to use weighted approaches in the interpretation/enrichment analyses.

- The authors use the pc function (t distribution) to estimate if the Pearson correlation coefficient is significant. This might not be accurate as the independence assumptions are likely violated. The authors could account for this by obtaining an empirical p-value based on a null distribution of correlation coefficients obtained from gene pairs not sharing miRNAs.

- If I understood correctly, the power law property is investigated by comparing the obtained SCOM network against random networks using a KS test on the path lengths. To my knowledge, this is unusual. A more common way to assess the power law property is to check if a power law can be fitted to the lower part of the node degree distribution as done by the poweRlaw R package, for example.

- The (multi-class) classification performance was compared against a baseline method but not against a random baseline - please repeat the classification after sample label (subtype label) permutation to see what a random baseline would look like.

- The classification results would be more convincing if applied to another data set, e.g. METABRIC for breast cancer.

- SCOMdb makes the results of this study easily accessible to users. However, the manuscript should also mention the state of the art, since there are numerous other web resources like SPONGEdb that SCOMdb should be compared against. https://doi.org/10.1002/wrna.1686 offers a review in Table 1 of such resources.

- The R code is offered as a download on the SCOMdb site but it would be much more convenient if the GitHub address was listed. The authors should also turn their code into an R package so it can be installed directly from GitHub via devtools.

# Minor

- The methods section is partially repetitive, e.g. the three criteria for SCOM are repeated several times.

- Page 12, bottom "when the influence of the synergistically competed mRNAs is eliminated". You mean miRNA here, not mRNA.

- The authors found the number of ceRNA interactions of pseudogene-related ceRNA networks to be significantly larger than that of lncRNA-related ceRNA networks. Not so surprising given that there are much more pseudogenes than lncRNA considered.

- It is not getting clear why the power law behavior is really important to consider. What does this change and mean?

- The authors might want to check out spongEffects in the future (https://doi.org/10.1093/bioinformatics/btad276). This method currently extracts modules from the ceRNA network by considering high-degree ceRNAs and their targets. SCOM delivers an alternative definition of a ceRNA module (a synergistic one) and its behavior across individual samples or subtypes might be better represented by spongEffect scores.

- The SCOMdb site is not accessible without the www prefix and does not use a secure connection. Both should be fixed.

- In SCOMdb, the search window is currently also a bit poor in features, e.g. result tables can not be sorted, filtered or downloaded. There are no visualizations (like bar charts, box or violin plots etc.) to illustrate the results. This should be improved in the future if the authors intend for their users to benefit from this page.

Reviewer #3: Due to its innovative creativity and well writing, I am willing to recommend that this manuscript be accepted and published. However, there are two issues that need to be explained in more detail by the author in the discussion: 1. Can the experimental path for verifying this theory be proposed in more detail in the discussion, rather than simply stating that further "experimental verification" is needed; 2. Regardless of the differences in tumor types (such as lung cancer, colorectal cancer, and/or pancreatic cancer), all types of tumor cells have common characteristics. Does the same SCOM exist in the systematic analysis at the pan cancer level in this study

**Have the authors made all data and (if applicable) computational code underlying the findings in their manuscript fully available?**

Reviewer #1: Yes

Reviewer #2: Yes

Reviewer #3: Yes

PLOS authors have the option to publish the peer review history of their article (what does this mean?). If published, this will include your full peer review and any attached files.

Reviewer #1: No

Reviewer #2: **Yes: **Markus List

Reviewer #3: No
---

## [Decision Letter · Decision Letter 1]

25 Sep 2023

Dear Dr. Zhang,

We are pleased to inform you that your manuscript 'Pan-cancer characterization of ncRNA synergistic competition uncovers potential carcinogenic biomarkers' has been provisionally accepted for publication in PLOS Computational Biology.

Best regards,

Mark Ziemann

Academic Editor

PLOS Computational Biology

Pedro Mendes

Section Editor

PLOS Computational Biology

Reviewer's Responses to Questions

**Comments to the Authors:**

Reviewer #1: Ther authors revised the manuscript following the reviewers' comments.

Reviewer #2: Thanks for addressing most of my comments. I would still urge the authors to consider empirical p values in the future in spite of the computational costs as the p values currently obtained may not reflect the true false positive rate well.

**Have the authors made all data and (if applicable) computational code underlying the findings in their manuscript fully available?**

Reviewer #1: Yes

Reviewer #2: Yes

PLOS authors have the option to publish the peer review history of their article (what does this mean?). If published, this will include your full peer review and any attached files.

Reviewer #1: No

Reviewer #2: **Yes: **Markus List

---

## [Editor Report · Acceptance letter]

3 Oct 2023

PCOMPBIOL-D-23-01031R1 

Pan-cancer characterization of ncRNA synergistic competition uncovers potential carcinogenic biomarkers

Dear Dr Zhang,

I am pleased to inform you that your manuscript has been formally accepted for publication in PLOS Computational Biology. Your manuscript is now with our production department and you will be notified of the publication date in due course.

With kind regards,

Anita Estes
